# Neural-PIL: Neural Pre-Integrated Lighting for Reflectance Decomposition

**Mark Boss**
University of Tübingen

**Varun Jampani**
Google Research

**Raphael Braun**
University of Tübingen

**Ce Liu***
Microsoft Azure AI

**Jonathan T. Barron**
Google Research

**Hendrik P. A. Lensch**
University of Tübingen

## Abstract

Decomposing a scene into its shape, reflectance and illumination is a fundamental problem in computer vision and graphics. Neural approaches such as NeRF have achieved remarkable success in view synthesis, but do not explicitly perform decomposition and instead operate exclusively on radiance (the product of reflectance and illumination). Extensions to NeRF, such as NeRD, can perform decomposition but struggle to accurately recover detailed illumination, thereby significantly limiting realism. We propose a novel reflectance decomposition network that can estimate shape, BRDF and per-image illumination given a set of object images captured under varying illumination. Our key technique is a novel illumination integration network called Neural-PIL that replaces a costly illumination integral operation in the rendering with a simple network query. In addition, we also learn deep low-dimensional priors on BRDF and illumination representations using novel smooth manifold auto-encoders. Our decompositions can result in considerably better BRDF and light estimates enabling more accurate novel view-synthesis and relighting compared to prior art. Project page: https://markboss.me/publication/2021-neural-pil/

## 1 Introduction

Inverse rendering is the task of decomposing a scene into its underlying physical properties, such as geometry and materials. Recovering these properties is useful for several vision and graphics applications such as view synthesis [10, 11, 51, 57], relighting [4, 10, 11, 23, 24, 38, 51, 58], and object insertion [7, 21, 38]. In this work, we aim to recover the 3D shape and spatially-varying bidirectional reflectance distribution function (SVBRDF) of an object imaged under different illumination conditions, as shown in Fig. 1. Estimating shape, illumination, and SVBRDF from 2D images is a highly ill-posed problem, as an observed pixel may appear dark either due to a dark surface material, or due to the incident light at that surface being reduced or absent.

Our approach follows the recent success of coordinate-based scene representation networks [14, 40, 43, 44, 46, 49] in representing 3D scenes for high-quality view-synthesis [44, 49]. These models decompose the scene into models of shape and radiance (emitted light), thereby enabling view synthesis. However, performing complete *inverse rendering* requires that radiance is decomposed further, into illumination and material appearances (SVBRDF) [6, 11, 51, 63]. A key component in learning these neural SVBRDF decomposition networks is the differentiable rendering [10, 11, 63] that generates images and gradients for the estimated lighting and SVBRDF parameters. These methods leverage traditional rendering techniques within modern deep learning frameworks to enable

---

*Work done while at Google.

35th Conference on Neural Information Processing Systems (NeurIPS 2021).

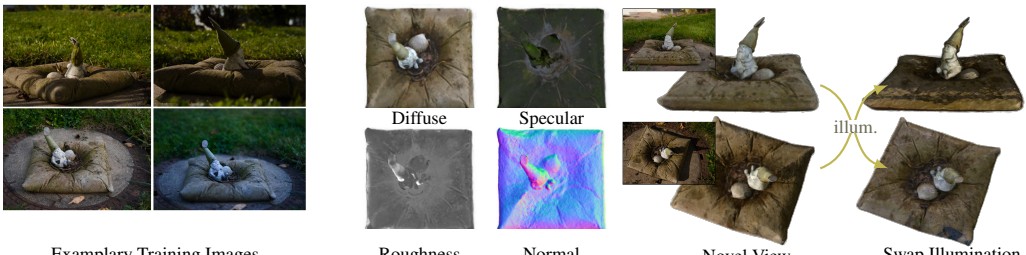

| Exemplary Training Images | Roughness | Normal | Novel View | Swap Illumination |

Figure 1: **Problem setting.** Our neural-PIL based technique decomposes images observed under unknown illumination into high-quality BRDF, shape and illuminations. This allows us to then synthesize novel views (targets shown in insets) and perform relighting or illumination transfer.

backpropagation. This is often expensive, as rendering requires computing integrals over the incoming light at each 3D location in the scene. As a remedy, recent works [11, 63] approximate the incident light by spherical Gaussians (SG), thereby accelerating the illumination integration. However, these SG representations lack the capacity required to model or recover the shape and material properties of highly reflective objects or images in complex natural environments.

In this work, we aim to replace the costly illumination integration step within these rendering approaches with a learned network. Inspired by the real-time graphics literature on image-based lighting [28], we propose a novel pre-integrated lighting (PIL) network that converts the illumination integration process used in rendering into a simple network query. Our neural-PIL takes as input a latent vector representation for the environment map, the surface roughness, and an incident ray direction, and from them predicts an integrated illumination estimate. This query-based approach for light integration results in efficient rendering and thereby simplifies and accelerates rendering and optimization. This neural light representation is also significantly more expressive than the more commonly used SG representation, thereby enabling higher-fidelity renderings. The architecture of our neural-PIL uses conditional multi-layer perceptrons (MLP) with FiLM layers [12]. Fig. 2 illustrates this illumination pre-integration for different surface roughness levels.

In addition, we also present a smooth manifold auto-encoder (SMAE), based on interpolating auto-encoders [5], that can learn effective low-dimensional representations of light and BRDFs. This learned low-dimensional space serves as a strong regularizer or prior for constraining the solution space of BRDFs and illumination. These constraints are critical, due to the ill-posedness of our problem setting. The smoothness of this manifold enables stable and effective gradient-based optimization of BRDF and light parameters. The neural-PIL, light-SMAE, and BRDF-SMAE networks are pre-trained on a dataset with high-quality environment maps (illumination) and materials (BRDFs). We integrate these component networks into our decomposition framework, in which we optimize a 3D neural volume with shape and SVBRDF while also optimizing per-image illuminations.

We perform an empirical analysis on synthetic datasets, along with qualitative and quantitative visual results on real-world datasets. We demonstrate that our decomposition network using our neural-PIL can estimate more accurate shape and material properties compared to prior art. The 3D assets with material properties produced by our model can be used to generate high-quality relighting and view-synthesis results with finer details compared to existing approaches.

## 2 Related Work

**Coordinate-based MLPs** allow spatial information to be stored within the weights of a neural network, thereby allowing the retrieval of information solely by querying coordinates [14, 43, 46, 52]. These methods have been combined with neural volume rendering [40] to enable photorealistic results on novel view synthesis, well-exemplified by NeRF [44]. In NeRF, a coordinate-based model is used to model a field of volumetric density and color, and renderings are produced by ray-marching through that neural volume. Though NeRF is capable of photorealistic renderings, it has many limitations that have been explored by recent work, such as: no relighting capabilities [6, 11, 42, 51, 63], long training times [39, 54], long inference times [11, 25, 31, 39, 60], extraction of 3D geometry and materials [11], and generalization [12, 54, 64]. This work addresses some of these challenges that enables relighting, extracts a conventional 3D geometry and material estimate, and enables real-time

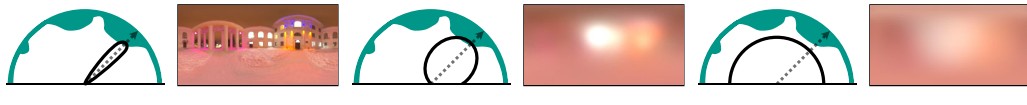

| Roughness 0.01 | Roughness 0.5 | Roughness 1.0 |

Figure 2: **Pre-integrated lighting.** As the roughness of the material increases, the reflected radiance depends on a larger region of the environment map. Brute-force integrals over the environment map are expensive, hence we propose a coordinate-based MLP that is trained to directly output the integrated illumination values conditioned on the surface roughness and view direction.

rendering (as our 3D assets are compatible with existing real-time rendering engines). The concurrent works of NeRD [11], NeRV [51] and PhySG [63] are most clostly related to ours. These methods decomposes the scene into shape and analytical SVBRDF parameters. However, NeRV requires known illumination, and NeRD and PhySG employ a spherical Gaussian (SG) model, which is not capable of modeling detailed illumination patterns.

**BRDF estimation** is a challenging research problem that aims to estimate the appearance of a physical material. For the highest accuracy results, measurements are performed under controlled laboratory conditions with known view and light positions [3, 9, 32, 33, 34], but this does not allow for the on-site capture of materials. Casual capture methods aim to solve this constraint by only requiring a camera, and sometimes a known light source. Often, machine learning techniques are leveraged to reduce the ambiguity through the use of data-drive priors and large datasets of BRDFs. Additional constraints from planar surfaces viewed under camera flash illumination are considered for single-shot [1, 16, 26, 36, 47], few-shot [1] or multi-shot [2, 9, 17, 18, 20] estimation. This casual setup can be extended to estimating the BRDF and shape of objects [6, 7, 8, 10, 30, 45, 47, 62] or scenes [38, 48]. Most of these methods are based on known active illumination. A limited number of light sources — most often a single one — are assumed to be responsible for the majority of illumination in a scene. Relying on only natural, uncontrolled illumination adds several additional challenges due to the drastically increased ambiguity across shape, illumination and BRDFs. Often these challenges are reduced by keeping the specular albedo non-spatially-varying, or by removing it entirely [35, 59, 63]. Other approaches require temporal traces and limit the casual capture setup [19, 56].

**Illumination estimation** from a single image is an inherently challenging problem. The task is inherently linked to BRDF estimation, as illumination affects appearance and is only indirectly observable from its interactions with surface materials. These two tasks are often solved in conjunction, sometimes by decomposing a single object into shape, reflectance, and a global set of spherical Gaussians (SGs) [11, 63]. Chen *et al.* [13] leverage a deep prior of environment maps with homogeneous materials, using an invertible neural BRDF model. Li *et al.* [38] decompose an entire scene into a simplified BRDF model with hemispherical SGs per point in the scene. The image of the environment in the background may be incorporated into prediction, shifting the problem to completion of the HDR environment map from sparse observations [21, 50, 55]. We not only learn a deep prior but a rendering aware network which is capable of integrating the environment illumination for a specific surface roughness enabling rendering the entire hemisphere of incoming light with a single evaluation.

## 3 Method

Given an image collection of an object captured under varying illumination conditions and camera viewpoints, we aim to jointly estimate the object's 3D shape and spatially-varying BRDF, as well as the illumination conditions of each image. Our input consists of a set of $q$ images with $s$ pixels each: $C_j \in \mathbb{R}^{s \times 3}; j \in \{1, \ldots, q\}$ along with per-pixel masks $M_j \in \{0, 1\}^{s \times 1}$ indicating which pixels belong to the object. Our goal is to learn a neural 3D volume $\mathcal{V}$ where, at each point $\boldsymbol{x} \in \mathbb{R}^3$, we estimate the BRDF parameters for the Cook-Torrance model [15] $\boldsymbol{b} \in \mathbb{R}^7$ (diffuse $\boldsymbol{b}_d \in \mathbb{R}^3$, specular $\boldsymbol{b}_s \in \mathbb{R}^3$, roughness $b_r \in \mathbb{R}$), unit-length surface normal $\boldsymbol{n} \in \mathbb{R}^3$ and optical density $\sigma \in \mathbb{R}$. In addition, we also estimate latent vectors representing per-image illumination $\boldsymbol{z}^l \in \mathbb{R}^{128}$. This problem statement corresponds to the practical application of recovering a 3D model of some

real-world object (for e.g., a statue or landmark) that has been photographed by different people at different times.

## 3.1 Preliminaries

**Brief overview of neural radiance fields (NeRF).** Our method is based on NeRF [44] which creates a neural volume for novel view synthesis using two Multi-Layer-Perceptrons (MLPs). The first MLP learns a coarse representation, which samples the 3D volume in a fixed sampling pattern, and the second MLP uses this knowledge to sample the volume in high-density areas at a finer resolution. The output of the MLP is a view-dependent color $c \in \mathbb{R}^3$ and optical density $\sigma \in \mathbb{R}$ for each given 3D location $x \in \mathbb{R}^3$ and view direction $d \in \mathbb{R}^3$. In order to render the output color $\hat{c} \in \mathbb{R}^3$ for a camera ray $r(t) = o + td$, with ray origin $o \in \mathbb{R}^3$ and view direction $d$, we approximate (via numerical quadrature) the integral $\hat{c}(r) = \int_{t_n}^{t_f} T(t)\sigma(t)c(t)\, dt$ with $T(t) = \exp(-\int_{t_n}^{t} \sigma(t)\, dt)$, using the near and far bounds of the ray $t_n$ and $t_f$ respectively [44].

**Image formation and image-based lighting.** NeRF directly models view-dependent color $\hat{c}$ at each 3D location. Thus, a simple image formation process that integrates the color information along camera rays is sufficient to render images. In contrast, we want to explicitly estimate an object material decomposition at each 3D location. We therefore must use a more explicit rendering formulation that relates image formation to BRDFs and illumination. The rendering equation [27] estimates the radiance $L_o \in \mathbb{R}^3$ at $x$ along the outgoing view direction $\omega_o \in \mathbb{R}^3$ ($\omega_o = -d$): $L_o(x, \omega_o) = \int_{\Omega} f_r(x, \omega_i, \omega_o; b) L_i(x, \omega_i)(\omega_i \cdot n)\, d\omega_i$, where $f_r$ is the BRDF evaluation, $L_i \in \mathbb{R}^3$ is incoming light, and $\omega_i \in \mathbb{R}^3$ is the incoming light direction. Using this single bounce rendering formulation, and ignoring exposure variation and tone-mapping, $L_o$ is equivalent to NeRF's color $\hat{c}$. The formulation can be split into diffuse and specular components:

$$L_o(x, \omega_o) = \underbrace{\frac{b_d}{\pi} \int_{\Omega} L_i(x, \omega_i)(\omega_i \cdot n)\, d\omega_i}_{\text{diffuse}} + \underbrace{\int_{\Omega} f_s(x, \omega_i, \omega_o; b_s, b_r) L_i(x, \omega_i)(\omega_i \cdot n)\, d\omega_i}_{\text{specular}} \quad (1)$$

where $f_s$ now only represents the specular portion of the BRDF evaluation. Following Karis *et al.* [28], several parts of this integration can be pre-computed [29]:

$$L_o(x, \omega_o) \approx \underbrace{(b_d/\pi)\tilde{L}_i(n, 1)}_{\text{diffuse}} + \underbrace{b_s(F_0(\omega_o, n)B_0(\omega_o \cdot n, b_r) + B_1(\omega_o \cdot n, b_r))\tilde{L}_i(\omega_r, b_r)}_{\text{specular}} \quad (2)$$

The illumination is now pre-integrated as: $\tilde{L}_i(\omega_r, b_r) = \int_{\Omega} D(b_r, \omega_i, \omega_r) L_i(x, \omega_i) d\omega_i$ which only depends on the mirrored view direction $\omega_r$ (which subsumes the surface normal $n$) and the roughness $b_r$, where $D$ describes the microfacet distribution [53]. This light pre-integration is illustrated in Fig. 2. Note that the same pre-integrated $\tilde{L}_i(\omega_r, b_r)$ is queried twice: the diffuse part captures the entire hemisphere and therefore is parameterized by the surface normal $n \in \mathbb{R}^3$ and a diffuse roughness of 1. The specular part looks up $\tilde{L}_i(\omega_r, b_r)$ for the reflected view direction $\omega_r \in \mathbb{R}^3$ and the specular roughness $b_r$. While the pre-integration already considers the microfacet distribution $D$, one must also account for shadowing, masking, and the Fresnel term. As shown in Karis *et al.* [28], these remaining parts can be pre-computed into two 2D lookup textures (LUT) $B_0$ and $B_1$ indexed by $(\omega_o \cdot n)$ and the roughness $b_r$. These are combined with the Fresnel term at normal incidence $F_0(\omega_o, n) = (1 - \omega_o \cdot h)^5$ with $h = \|\omega_i + \omega_o\|$.

This pre-integration approach replaces the complex integration during shading with a set of simple additions and multiplications. We have integrated the core idea of this approach into an efficient differentiable neural rendering framework, which allows for the optimization of geometry, BRDF, and illumination simultaneously via standard backpropagation. We further reduce the computational complexity by mimicking the pre-integration of $\tilde{L}_i(\omega_r, b_r)$ with a simple query through our Neural-PIL that operates directly on a neural representation of the illumination, as we will now explain.

## 3.2 Decomposition with Neural-PIL

Fig. 3 shows the neural decomposition architecture which closely follows the architectures of NeRF [44] and NeRD [11], but with some key differences. The coarse network learns a view and illumination-dependent color whereas the fine network decomposes the scene into BRDF parameters.

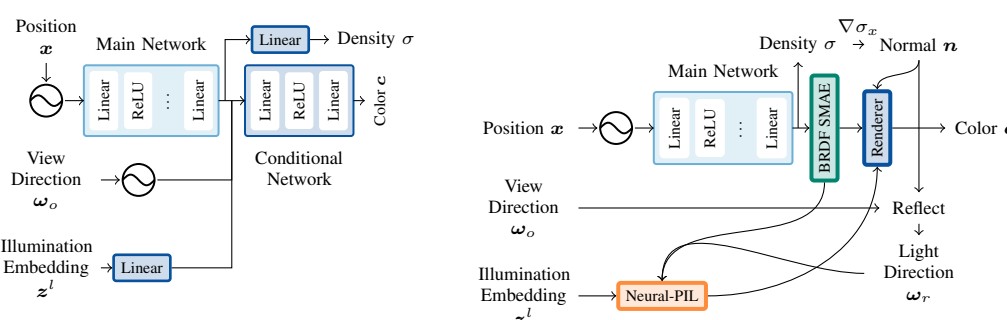

Figure 3: **Decomposition with Neural-PIL architecture.** (a) Similar to NeRF-W[42] our coarse network uses a latent illumination estimate to predict a view-dependent color and density. (b) Pre-trained networks restrict the possible BRDF representation (BRDF-SMAE) and the incident lighting (PIL) to lower-dimensional spaces. A single evaluation of Neural-PIL returns a pre-filtered illumination cone according to surface roughness. Using that, the BRDF estimate, and a surface normal (the unit-norm gradient of our density estimate), we render the shaded color $c$.

**Coarse network.** Like in NeRF [44], the aim of the coarse network is to obtain rough point density that helps in finer sampling for the following decomposition network. As illustrated in Figure 3a, the coarse network takes 3D location $x$, view direction $\omega_o$ and illumination embedding $z^l$ as input and predicts point density $\sigma$ and color $c$ at $x$. In contrast to NeRF, which estimates view-conditioned colors, we estimate both view and illumination conditioned colors, as our input images can be captured under varying illumination. Refer to the supplement for architecture details.

**Decomposition network.** The decomposition network estimates density $\sigma$ and BRDF embedding $z^b \in \mathbb{R}^4$ at each 3D location $x$ in the implicit volume. As illustrated in Figure 3b, the conditional network in the coarse network is replaced by explicit rendering in the decomposition network. There are two key innovations in the decomposition network: 1) Use a novel pre-integrated light (PIL) network that results in efficient rendering while also representing the illumination with high fidelity. 2) We learn smooth low-dimensional manifolds to represent illumination and BRDF parameters, which serve as strong priors. We will now explain our rendering process, the Neural-PIL, and the smooth manifold auto-encoders (SMAE).

**Rendering process.** For rendering, we use the rendering formulation in Equation 2. We estimate normal $n$ at $x$ by computing the gradient $\nabla\sigma_x$ of the density w.r.t. the input position (by passing gradients through the decomposition network). We also convert the BRDF embedding $z^b$ into BRDF parameters $b$ with our BRDF-SMAE. Unlike NeRF [44], which integrates sample colors along the camera ray, we first compute the expected termination of each ray (similar to depth map) with the sample densities along the camera ray, and then do the rendering only at that point along each camera ray. At the ray termination positions, we compute the integrated illumination $\tilde{L}_i(\omega_r, b_r)$ and use Equation 2 for rendering with the estimated BRDF $b$ and normal $n$.

**Neural-PIL.** The integration of the incoming light is traditionally approximated by Monte Carlo sampling, in which illumination contributions from many directions are numerically integrated. The computation of the pre-integrated $\tilde{L}_i(\omega_r, b_r)$ also involves either this costly numerical accumulation, or a convolution performed on the complete environment map — though neither approach is practical within a differential rendering engine. We therefore learn a network that performs this light pre-integration, thereby converting the costly integral computation into a simple network query. The architecture of the Neural-PIL is visualized in Fig. 4. The Neural-PIL takes as input the illumination embedding $z^l$, the incoming light direction $\omega_r$ and the roughness $b_r$ at a point, and directly predicts the pre-integrated light $\tilde{L}_i(\omega_r, b_r)$. The aim of the Neural-PIL is to first decode the illumination along the incoming mirror direction $\omega_r$ from the given embedding $z^l$ and then mimic the light pre-integration process for the surface roughness $b_r$. Following this general intuition, the Neural-PIL takes $\omega_r$ as input, and we condition the first few layers of the network with illumination $z^l$, and condition a later layer with roughness $b_r$. The first few layers are intended to decode all required

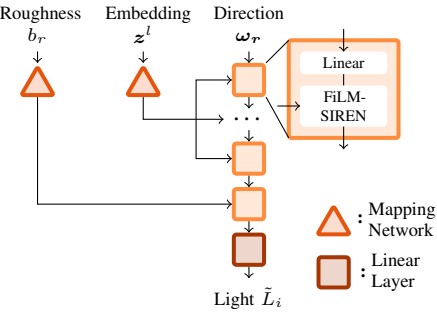

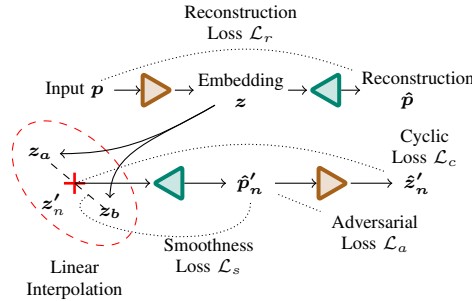

Figure 4: **Neural-PIL.** A coordinate-based MLP returns the pre-integrated radiance for the query direction, where roughness determines the integration footprint.

Figure 5: **Smooth manifold auto-encoder.** By imposing specific losses on interpolations between input samples, our Smooth Manifold Autoencoder encourages a smooth embedding space.

illumination information for the given direction, and the later layers are intended to perform light integration conditioned on the material roughness.

For the Neural-PIL design, we leverage a pi-GAN-like [12] architecture with FiLM-SIREN layers. Each FILM-SIREN layer [12] takes the modulating parameters (a scalar $\lambda_0$ and two vectors $\boldsymbol{\beta}$ and $\boldsymbol{\gamma}$) to modulate the output $\boldsymbol{y}$ of the earlier linear layer followed by sine computation as follows: $\phi(\boldsymbol{y}) = \sin(\lambda_0 \boldsymbol{\gamma} \odot \boldsymbol{y} + \boldsymbol{\beta})$, where $\odot$ denotes a Hadamard product. In our Neural-PIL, we employ two mapping networks to predict modulating parameters$(\boldsymbol{\gamma}, \boldsymbol{\beta})$ for the FiLM-SIREN layers. The first mapping network generates the modulating parameters for the first layers from the illumination embedding $\boldsymbol{z}^l$, while the second one for the penultimate layer from the given roughness $\boldsymbol{b}_r$.

In addition to converting a costly light integration process in the rendering into a simple network query, our Neural-PIL has another key advantage. State-of-the-art differentiable rendering frameworks [11, 22, 38, 63] that work with BRDFs use either spherical Gaussian (SG) or spherical harmonic (SH) light representations, which both suffer from lack of fine details. Although one could represent fine illumination details with a large number of SG or SH bands, this parameter increase would also make the rendering prohibitively slow with high memory costs. In contrast, our Neural-PIL is an MLP that directly produces pre-integrated light required for the rendering. Our experiments also demonstrate that our Neural-PIL can represent finer details in illumination compared to SG representation (Sec. 4 - Fig. 6). See the supplementary for more Neural-PIL architecture details.

**Smooth manifold auto-encoder (SMAE).** Since jointly estimating 3D shape, materials and lighting is a highly underconstrained problem, in order to converge to plausible solutions, we must regularize optimization towards likely illuminations and BRDFs. For this, we learn low-dimensional smooth manifolds that capture the data distribution of BRDFs and illuminations. In addition to acting as strong priors, optimizing on smooth manifolds (as opposed to directly optimizing on standard BRDF space, which need not be smooth) allows for more effective gradient-based optimization of reflectance decomposition for a given scene. Fig. 5 illustrates our SMAE that we use to learn separate low-dimensional manifolds to represent BRDF and illumination embeddings. Specifically, we use Interpolating Autoencoders [5] with several additional loss functions. The encoder network $E$ takes input $\boldsymbol{p}$ (either BRDF or light environment map) and generates the latent embedding $\boldsymbol{z}$, which is then passed onto the decoder network $G$ that generates an input reconstruction $\boldsymbol{p}'$. We then randomly sample two latent vectors from the mini-batch: $\boldsymbol{z}_a$ and $\boldsymbol{z}_b$; followed by sampling $m \in \mathbb{N}$ linearly interpolated embeddings that are uniformly spaced between $\boldsymbol{z}_a$ and $\boldsymbol{z}_b$: $\{\boldsymbol{z}'_n\}|n = 1, 2, \ldots, m$. We pass each of these interpolated latents $\boldsymbol{z}'_n$ through the decoder $G$ and the encoder $E$ to obtain $\hat{\boldsymbol{p}}'_n$ and $\hat{\boldsymbol{z}}'_n$, respectively. Using the four losses depicted in Fig. 5 the encoder and decoder networks are trained jointly. One is the standard reconstruction loss $\mathcal{L}_r$ between input $\boldsymbol{p}$ and reconstruction $\boldsymbol{p}'$. In addition, we add a discriminator network on $\hat{\boldsymbol{p}}'_n$ and use the standard adversarial loss $\mathcal{L}_a$ used in LSGAN [41], which ensures that the interpolated latent vectors can generate plausible data. We enforce a bijective mapping of the encoder and the decoder with a cyclic loss $\mathcal{L}_c$ which is the $L_2$-loss between the interpolated latents $\{\boldsymbol{z}'_n\}$ and their re-estimated counterparts $\{\hat{\boldsymbol{z}}'_n\}$. Lastly, to ensure that the learned embedding space is smooth, we impose a smoothness loss $\mathcal{L}_s$ on the gradient of decoder

$G$ w.r.t. the interpolating scalar value $\alpha$: $\mathcal{L}_s = 1/m \sum_n (\nabla_\alpha G(z'_n))^2$. The total loss to train SMAE is a combination of the 4 losses: $\mathcal{L} = \mathcal{L}_r + \lambda_1 \mathcal{L}_a + \lambda_2 \mathcal{L}_c + \lambda_3 \mathcal{L}_s$.

Despite being only 7-dimensional, the space of the Cook-Torrence BRDF representation [15] that we use is too unconstrained for our task, and imposes strong correlations between the diffuse and specular terms of real-world materials. We therefore train a BRDF-SMAE with an MLP encoder and decoder that maps these 7D parameters into 4D latent embeddings $z^b \in \mathbb{R}^4$ using a dataset of real-world BRDF material collections [10]. Similarly, real-world illuminations exhibit significant statistical regularities: lights are more likely to be tinted blue or yellow, and brighter light is more likely to coming from above than below. To capture this regularity, we train a Light-SMAE with CNN encoder and decoder on a dataset of 320 environment maps from [61]. We then map the $128 \times 256$ 2D environment maps onto a 128-dimensional smooth latent space, $z^l \in \mathbb{R}^{128}$. We provide more network and training details for BRDF-SMAE and Light-SMAE in the supplementary.

**Training.** Since we have several networks in our decomposition learning pipeline, we will briefly explain the overall training protocol here with more details in the supplementary. We first train Light-SMAE and BRDF-SMAE with a dataset of environment maps and BRDFs respectively. The Neural-PIL network is then trained with the manifold created by the frozen Light-SAME encoder. This separation is mainly done to ease the memory requirements of training both networks jointly. With the frozen BRDF-SMAE's decoder in the decomposition network and with Neural-PIL in the rendering step, we jointly optimize both the coarse network and the decomposition network for a given set of scene images. For stability, we only optimize the illumination embedding $z^l$ via decomposition network and we do not backpropagate the loss signal onto illumination in the coarse network. More training details can be found in the supplement.

## 4 Experiments

We evaluate our approach w.r.t. different baselines on the aspects of BRDF and light estimation, view synthesis, and relighting.

**Baselines.** The closest work to ours is NeRD [11] which forms our primary comparison across different evaluations. To our knowledge, there exists no other published work that tackles the same problem of estimating shape, illumination and BRDF from images of varying illumination. For view synthesis, we also compare with NeRF [44]. For BRDF evaluations, we also compare with Li *et al*. [37] which does BRDF decomposition from a single image. In addition, we combine Li *et al*. [37] with NeRF [44] to create a baseline that is closer to our problem setting.

**Datasets.** To enable the comparisons with NeRD [11], we use the publicly released dataset used in [11] which provides 3 synthetic (Chair, Globe, Car) and 4 real-world

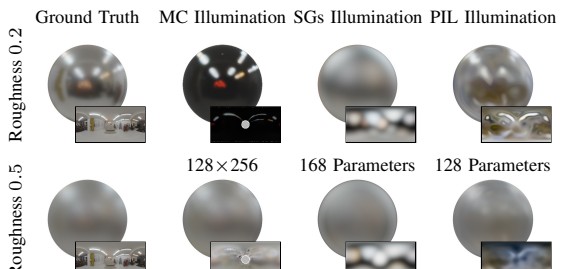

Figure 6: **Neural-PIL *vs*. spherical Gaussian (SG) *vs*. Monte-Carlo (MC) integration renderings.** With known geometry and reflectance, we optimize using MC integration for the direct illumination, SGs as well as our latent illumination via Neural-PIL. This figure shows final renderings with the optimized light parameters, while the recovered illumination is shown in the insets. Despite Neural-PIL having fewer parameters, it is able to recover more detailed environment maps and thereby produce accurate renderings.

scenes. Two real-world datasets consist of multi-view captures with fixed, unknown illumination (Cape), relatively varying illumination (Head) and two others where the illumination varies with each image (Gnome, MotherChild). In addition, we also present view synthesis results on datasets (Ship, Chair, Lego) used in NeRF [44].

**Fildelity of Neural-PIL.** Since Neural-PIL forms the key component of our decomposition framework, we evaluate its learned light representation against a more commonly used spherical Gaussians (SG) representation. Additionally, we add a baseline which directly optimizes an environment map using Monte-Carlo (MC) integration. We render a simple metallic sphere using an unseen environment as shown in Fig. 6, with two different roughness levels $0.2$ and $0.5$. Assuming known roughness

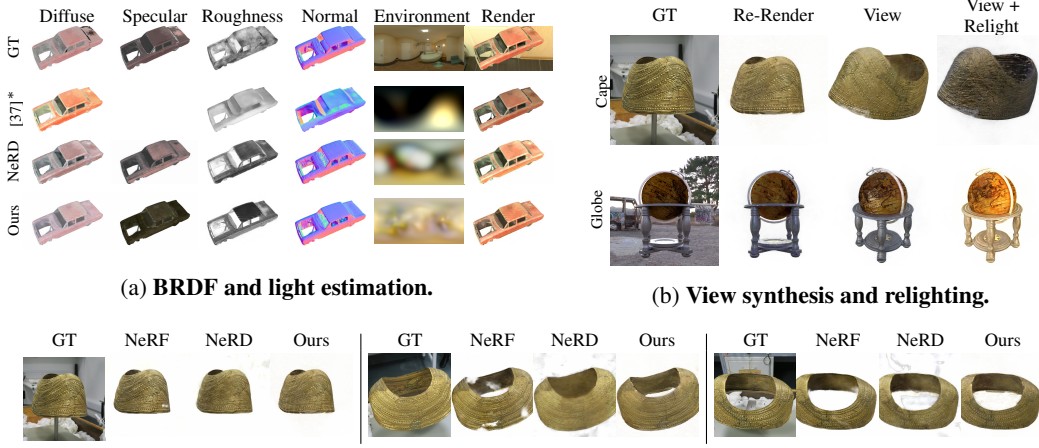

(a) **BRDF and light estimation.**

(b) **View synthesis and relighting.**

(c) **View synthesis with NeRF, NeRD and our approach.**

Figure 7: **Visual comparisons.** (a) Our model produces more accurate BRDF and illumination estimates, which results in more faithful rendering results. (b) To evaluate view synthesis and relighting we keep the camera and light fixed (col 2), then move the camera (col 3), and then adjust the lighting (col 4). (c) Even when using a single illumination (the problem setting used by NeRF) our method produces shape estimates with fewer artifacts and more detail than both NeRF or NeRD.

and shape, we optimize for SG illumination using the SG-based differentiable rendering used in NeRD [11]. Similarly, we optimize the latent illumination representation using our Neural-PIL-based renderer. For the MC baseline, we leverage BRDF importance sampling, which based on the surface roughness describes how the rays would likely scatter. Here, we cast 128 samples-per-pixel (spp) based on the BRDF towards the environment map with a resolution of $128 \times 256$. The resulting estimated MC, SG illumination and Neural-PIL illuminations are shows in Fig. 6. Compared to the SG illumination model with 24 lobes and 168 parameters, our recovered illumination vector $z^l$ with only 128 dimensions captures more details, especially in the high-frequency light panels. This leads to a significantly reduced rendering error for both roughness values even though the illumination prediction is more ambiguous for rougher materials. While the MC integration could easily recover detailed highlights, the remaining areas are not recovered well. Besides the improved quality, Neural-PIL based rendering is also much faster. Rendering million samples with our Neural-PIL network takes just $1.86 \, \mathrm{ms}$ compared to $210 \, \mathrm{ms}$ rendering with 24 SGs. Table 1 shows average PSNR on 6 rendered spheres with more visual results similar to Fig. 6 in the supplements. Our method outperforms both baselines in reconstruction quality.

**Ablation study.** To showcase the effectiveness of our novel additions, we perform an ablation of the BRDF-SMAE. Table 2 shows the influence of the BRDF-SMAE on material estimation. These are the PSNR values on the 3 synthetic scenes under varying illumination. It is clear from the table that, especially in estimating the specular parameter, using BRDF-SMAE improves the results drastically. As this parameter is also tied to the diffuse color a degradation in performance is expected. The roughness parameter – even though it is uncorrelated to diffuse and specular – is also improved most likely due to improved color parameters. For the ablation of Neural PIL network, one can refer to NeRD [11] as a baseline that neither uses BRDF-SMAE nor Neural PIL. PSNR metrics in Table 3a shows that our method can result in better decomposition compared to [11].

**BRDF evaluations.** Following the results in NeRD [11], Table 3a shows the BRDF estimation metrics for different techniques computed on the scenes Globe, Car and Chair. When compared with NeRD, our approach resulted in better diffuse and roughness parameters. Only the prediction of the specular parameter is worse compared to NeRD. This may be due to NeRD's basecolor-metallic parameterization, which can reduce some ambiguity but also limits the space of expressible materials. A visual comparison is shown in Fig. 7a demonstrating clear visual improvements w.r.t. [37]. One can observe higher frequency details in the environment map using our approach compared to NeRD and the final renderings also show that our result is closer to GT rendering (top-right). Refer to the supplementary material for more visual results.

| Roughness | MC | SGs | Neural-PIL (Ours) |
|---|---|---|---|
| 0.2 | 34.88 | 31.57 | **35.76** |
| 0.5 | 35.14 | 28.98 | **35.28** |

Table 1: **Better illumination estimates with Neural-PIL.** Average PSNR with 6 rendered spheres shows that Neural-PIL achieves better PSNR over the spherical Gaussian (SG) and Monte-Carlo integration (MC) baselines. More accurate illuminations also enables improved BRDF decomposition and relighting.

| Parameter | w/o BRDF SMAE | Ours |
|---|---|---|
| Diffuse | 11.87 | **20.22** |
| Specular | 9.24 | **16.84** |
| Roughness | 16.51 | **24.82** |

Table 2: **Ablation study.** Average PSNR of BRDF estimation on 3 synthetic scens under varying illumination demonstrates the positive influence of using the BRDF-SMAE to constrain the BRDF parameter space.

(a) **BRDF decomposition**

| PSNR↑ | [37] | [37]+[44] | [11] | Ours |
|---|---|---|---|---|
| Diffuse | 1.06 | 1.15 | 18.24 | **20.22** |
| Specular | — | — | **25.70** | 16.84 |
| Roughness | 17.18 | 17.28 | 15.00 | **24.82** |

(b) **View synthesis**

| | Synthetic | | Real-World | |
|---|---|---|---|---|
| Method | PSNR↑ | SSIM↑ | PSNR↑ | SSIM↑ |
| NeRF | **34.24** | **0.97** | 23.34 | 0.85 |
| NeRD | 30.07 | 0.95 | 23.86 | 0.88 |
| Ours | 30.08 | 0.95 | **23.95** | **0.90** |

(c) **View synthesis and relighting**

| | Synthetic | | Real-World | |
|---|---|---|---|---|
| Method | PSNR↑ | SSIM↑ | PSNR↑ | SSIM↑ |
| NeRF | 21.05 | 0.89 | 20.11 | 0.87 |
| NeRD | 27.96 | 0.95 | 25.81 | **0.95** |
| Ours | **29.24** | **0.96** | **26.23** | **0.95** |

Table 3: **Comparisons with baselines.** (a) A comparison against methods for BRDF decomposition under unknown illuminations, where we see that our model performs well (consistent with our improved relighting performance). (b) An evaluation of view-synthesis (without relighting) under a single illumination, where our model performs well despite this not being our primary task. (c) Here, input images are taken under different illumination conditions, so joint relighting and view synthesis are required. Our model outperforms both baselines by a significant margin: NeRF (which is not intended to address this task) but also NeRD (which targets this same problem statement).

**View synthesis and relighting.** On the datasets with fixed illumination (Cape, NeRF-Ship, NeRF-Chair, NeRF-Lego), we can directly compare our renderings with existing novel view synthesis techniques (both NeRF [44] and NeRD [11] here). Table 3b shows novel view evaluation metrics on these datasets with fixed illumination. Results show that our results are better than NeRD showing the improved capture of view-dependent effects. NeRF still outperforms NeRD and our method in the synthetic fixed illumination setting, but is outperformed on the real-world fixed illumination dataset. However, the fixed illumination might in general limit the decomposition capabilities, as shadows do appear always at the same surface locations and therefore might not be correctly disentangled from the BRDF.

On datasets with varying illumination across images (Gnome, MotherChild, Chair, Car, Globe, Head), we need to do both view synthesis and relighting to generate novel unseen test views. Table 3c displays the results on these datasets. NeRF [44] can not do relighting and is included as a weak baseline. The results are significantly better than NeRD and shows that our method can more faithfully estimate the underlying parameters resulting in better relighting under novel illumination conditions.

Fig. 7b shows a couple of results with view synthesis and relighting. The renderings demonstrate realistic view synthesis including re-lighting. Fig. 7c shows novel view synthesis comparison with NeRF and NeRD on Cape scene captured with fixed illumination. Despite NeRF being a strong baseline it could not recover the complete surface due to the reflectiveness. On the other hand, our view synthesis results are more close to the GT on unseen views compared to both NeRF and NeRD.

## 5 Conclusion

We presented a novel reflectance decomposition technique that can estimate shape, per-image illumination and BRDF from images captured in unknown and varying illuminations. The key innovation is the neural-PIL network that can replace costly light integration during rendering with a simple network query resulting in a fast and practical differentiable rendering with high-fidelity illumination. In addition, we propose novel learning techniques with SMAEs that can learn effective low-dimensional smooth manifolds for both BRDF and light representations. Experiments on both synthetic and real-world scenes demonstrate superior decomposition results along with better novel view synthesis and relighting in comparison to prior art.

**Limitations.** While our techniques make significant strides in the areas of differentiable rendering as well as shape and material decomposition, several challenges still remain in this complex problem setting. Our approach can not handle inter-reflections. Concurrent works such as NeRV [51] are capable of handling inter-reflections and shadowing but only for known illumination. Due to the large ambiguity between the interplay of all effects, solving everything jointly is an extremely challenging problem. Another limitation of our method is that we can not guarantee to converge to the correct underlying BRDF, reflectance and illumination. Our loss is only photometric and, therefore, we find one solution which explains all input images, *e.g.*, adding a new input image might converge to a different representation, as new effects are visible. Also, while our neural-PIL network is capable of producing higher frequency illumination with fewer parameters compared to standard representations such as SGs, mirror-like reflections are still not possible and therefore can limit the reconstruction quality when mirror-like surfaces are present in the scene.

**Broader impact.** As is generally the case in machine learning, biases in the data used during training may result in biases in the learned model. Our pre-trained networks for BRDFs and incident illumination serve as priors on materials and lighting conditions, and so any bias in the training data used for pre-training those models may result in bias in our estimations of materials and illumination. If the presented techniques were applied to human subjects (which we do not do here) the performance of the model might vary as a function of the subject's skin color for skewed training distributions.

The purpose of our model is to better enable the creation of highly accurate 3D models from photographs, which could then be used as visual effects in film or television, or in video games. Currently, the creation or acquisition of 3D assets is largely the domain of specialized CGI artists. Improved tools for automating this task may lower the barrier to entry into these careers, which may be seen as harming job opportunities for artists already working in this area. Despite this, we are hopeful that the commoditization of tools for 3D model acquisition will have a net positive impact by allowing a wider range of people to automatically construct high-fidelity 3D models from their image collections.

## Acknowledgments and Disclosure of Funding

This work has been funded by the Deutsche Forschungsgemeinschaft (DFG, German Research Foundation) under Germany's Excellence Strategy – EXC number 2064/1 – Project number 390727645 and SFB 1233, TP 02 - Project number 276693517. It was supported by the German Federal Ministry of Education and Research (BMBF): Tübingen AI Center, FKZ: 01IS18039A.

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
