*Supplementary Material for*
# Neural-PIL: Neural Pre-Integrated Lighting for Reflectance Decomposition

**Mark Boss**
University of Tübingen

**Varun Jampani**
Google Research

**Raphael Braun**
University of Tübingen

**Ce Liu**[*]
Microsoft Azure AI

**Jonathan T. Barron**
Google Research

**Hendrik P. A. Lensch**
University of Tübingen

This supplementary provides additional training details as well as further results from our method.

## A1   Architecture and training details

**BRDF-SMAE.** In our decomposition network, a specific BRDF embedding is stored in a neural volume at each point. Our BRDF-SMAE should therefore be able to encode a single BRDF. As each point in the neural volume can have a different embedding, the resulting decomposition has a spatially varying BRDF. To encode this singular BRDF per point, we leverage a MLP network architecture. For the encoder, decoder, and discriminator, we use 3 MLP layers with 32 output features. We train our SMAE to create a smooth latent space on the material dataset of Boss *et al.* [1]. We set the SMAE loss weighting to $\lambda_1 = 0.01$, $\lambda_2 = 0.01$ and $\lambda_3 = 0.001$. We use 64 interpolation steps in the latent space and use a mean absolute error (MAE) on the BRDF parameters for the reconstruction loss. In total, we use 1.5 million training steps with a batch size of 256 for training on a single NVIDIA 1080 TI GPU. This roughly takes 3.5 hours to converge. We use the Adam optimizer with a learning rate of 1e-4.

**Light-SMAE.** As our dataset only consists of 320 environment maps, we augment the dataset by randomly rotating each environment map 10 times, and during training, we randomly blend two environment maps. Additionally, we downscale the environment maps to $128 \times 256$. We set the SMAE loss weighting to $\lambda_1 = 0.01$, $\lambda_2 = 0.0001$ and $\lambda_3 = 0.05$ with 5 interpolation steps between each of the batch halves. Due to the high dynamic range, we found that specific care is required to ensure smooth training. The input to the encoder is transformed from HDR to LDR by $\log(1 + x)$ and the output from LDR to HDR with $\exp(x - 1)$. We further calculate the loss on a logarithmic scale using the MALE loss: $|\log(1 + x^*) - \log(1 + \hat{x})|$.

Our networks are all based on CNNs, whereas the encoder and discriminator leverage CoordConvs [6]. The encoder and discriminator do not use padding, whereas the decoder uses the "same" padding. The overall architecture is shown in Table A1. In total, we use 4 million steps with a batch size of 24 to train on a single NVIDIA 1080 TI GPU, which takes 5 days to train. The Adam optimizer with a learning rate of 5e-4 is used for the training.

**Neural-PIL.** We train the neural-PIL using the same environment maps dataset as used for training the Light-SMAE. Here, the encoder of the Light-SMAE is used for defining the smooth latent space. The network is comprised of MLPs with the FiLM-SIREN conditioning [3]. The first portion of the network is comprised of 3 layers with 128 features. The $\beta$ and $\gamma$ conditioning parameters are generators from a mapping network with 2 MLPs with 128 elu activated features. An additional MLP output layer produces 768 features, which corresponds to 128 $\beta$ and 128 $\gamma$ features per layer.

---

[*]Work done while at Google.

35th Conference on Neural Information Processing Systems (NeurIPS 2021).

### (a) Encoder

| Type | Size | Stride | Features | Activation |
|---|---|---|---|---|
| CoordConv | 3 | 1 | 8 | elu |
| CoordConv | 4 | 2 | 21 | elu |
| CoordConv | 3 | 1 | 21 | elu |
| CoordConv | 4 | 2 | 42 | elu |
| CoordConv | 3 | 1 | 42 | elu |
| CoordConv | 4 | 2 | 64 | elu |
| CoordConv | 3 | 1 | 64 | elu |
| Flatten |  |  |  |  |
| MLP |  |  | 128 | Linear |

### (b) Decoder

| Type | Size | Stride | Features | Activation |
|---|---|---|---|---|
| ConvT | (1,2) | 1 | 64 | elu |
| ConvT | 4 | 2 | 58 | elu |
| Conv | 3 | 1 | 58 | elu |
| ConvT | 4 | 2 | 52 | elu |
| Conv | 3 | 1 | 52 | elu |
| ConvT | 4 | 2 | 45 | elu |
| Conv | 3 | 1 | 45 | elu |
| ConvT | 4 | 2 | 39 | elu |
| Conv | 3 | 1 | 39 | elu |
| ConvT | 4 | 2 | 32 | elu |
| Conv | 3 | 1 | 32 | elu |
| ConvT | 4 | 2 | 26 | elu |
| Conv | 3 | 1 | 26 | elu |
| ConvT | 4 | 2 | 20 | elu |
| Conv | 3 | 1 | 20 | elu |
| Conv | 1 | 1 | 3 | $\exp(x-1)$ |

### (c) Discriminator

| Type | Size | Stride | Features | Activation |
|---|---|---|---|---|
| CoordConv | 3 | 1 | 8 | relu |
| CoordConv | 4 | 2 | 32 | relu |
| Conv | 3 | 1 | 32 | relu |
| CoordConv | 4 | 2 | 32 | relu |
| Conv | 3 | 1 | 32 | relu |
| CoordConv | 4 | 2 | 32 | relu |
| Conv | 3 | 1 | 32 | relu |
| CoordConv | 4 | 2 | 32 | relu |
| Conv | 3 | 1 | 32 | relu |
| Flatten |  |  |  |  |
| MLP |  |  | 1 | Linear |

Table A1: **Light-SMAE Architecture.** Details for the architecture used for each network. Conv denotes a regular 2D conv, ConvT a transposed 2D convolution and CoordConv uses a 2D convolution with the coordinates as described in [6].

The penultimate layer is then conditioned on the roughness $b_r$, which is parametrized by a mapping network with the first elu activated MLP outputting 32 features and the second 256 for the $\beta$ and $\gamma$ mapping parameters of the respective layer. Finally, a final MLP in the main network generates the final output color with 3 features output and $\exp(x - 1)$ as the activation to generate easier HDR values.

For training, we first try to encode the full environment map with a roughness of 0. In addition, we perform reconstruction with 8192 random directions and roughness levels. Both perform both of these simultaneously with a batch size of 8. In total, we use 4 million steps to train on a single NVIDIA 1080 TI GPU, which takes 4.5 days to train. We use the Adam optimizer with a learning rate of 5e-4.

**Decomposition** Both the coarse and fine networks consist of 8 MLPs with 256 relu activated features each. We randomly sample 4096 ray directions per image for training. The ray directions are also jittered as in Boss *et al.* [2]. For real-world data, we sample the ray in disparity rather than linear in depth. This places more samples close to the camera. In total, we train 400,000 steps on 4 NVIDIA 2080 TI GPU, which takes about 22 hours. We use an adam optimizer with a learning rate of 4e-4.

We employ additional exponentially decaying losses over 10,000 steps. We use the background segmentation loss, similar to [2], which ensures rays that do not hit the object do not contribute and additionally add a BRDF priming loss. This loss initially sets the diffuse color to the actual image color and the roughness to $0.3$ using a Mean Squared Error (MSE). The background segmentation loss fades in over the duration, and the BRDF priming loss fades out. Our main reconstruction loss is an MSE between the rendered color $c$ and the corresponding pixel in the input image. This loss is then exponentially faded over 100,000 steps to a cosine weighted MSE: $(\boldsymbol{x}^* \, \boldsymbol{\omega}_o \cdot \boldsymbol{n} - \hat{\boldsymbol{x}} \, \boldsymbol{\omega}_o \cdot \boldsymbol{n})^2$. This weighting tends to achieve better BRDF fitting results [4] as harsh grazing highlights from the Fresnel effect are not factored as much as regular samples, as well as our approximated rendering model being the least accurate in the grazing angles. The reason for this fading loss scheme is that the normals $\boldsymbol{n}$ are not reliable in the early stages of the training.

## A2 SMAE ablation study.

The main goal of the SMAE is to enable optimizing a latent space from backpropagation through the decoder alone. In Fig. A2, we show the estimated BRDF parameter maps (unseen taken from `www.sharetextures.com`) by backpropagation through the decoder for 200 steps with the Adam

optimizer and a learning rate of $0.01$. Additionally, we show the smoothness of the latent space by interpolating 4 materials in a grid. If the loss is working as intended, the transition between each corner of the 3 parameter maps should be smooth. As seen in Fig. A2, only with all losses active, we can successfully optimize the materials Small artifacts remain. For example, the joints in the tile have become metallic as the method correctly learned the constraint of a black diffuse color often indicates a metal. However, the space is smooth and allows for adding deep priors with a smooth gradient-based optimization.

## A3  Results

**Neural-PIL *vs*. SGs. *vs*. MC.** For the results in Fig. 6 of the paper, the SGs or the latent representation for the Neural-PIL was optimized from known reflectance and shape, respectively. In Fig. A1, we show more results of the same optimizations with different environment maps. We use a rendered image of a metallic sphere with roughness values of $0.2$ and $0.5$ as the targets. We then optimize the illumination with 1000 steps such that the SGs Renderer, MC Renderer or the PIL rendering respectively match the target as well as possible. The environments shown for the PIL illumination are the pre-integrated maps that encode the blur caused by the roughness, while the SGs environments are again convolved with the BRDF lobe to obtain the rendered image of the sphere. For the MC renderer the environment maps are shown as optimized. Notice the high-frequency detail captured with Neural-PIL. *E.g*. the gap between the buildings is captured accurately in the top right.

**BRDF decomposition.** Fig. A3 shows more BRDF decomposition results where we compare our results with GT, NeRD [2] and Li *et al*. [5]. Notice the accurate relighting results.

**Visual comparison on all scenes** Fig. A4 shows the quality of the novel view synthesis and also novel relighting – for Gnome and Mother-Child – on every scenes with our baseline. As seen our method provides convincing results over all our test scenes. The lego scene exhibits a slight color shift in our method. This is due to this scene being captured under an artificial illumination setup which is in complete darkness, except of two large white area lights. This is hard to reproduce for our manifold of natural illuminations. The BRDF is also constrained by natural materials and is therefore not capable of adjusting for the illumination.

## A4  Dataset licenses

The environment maps are taken from `hdrihaven.com` which is under the CC0 license. BRDFs are extracted from the Boss *et al*. [1] which is released under the NVIDIA Source Code License.

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

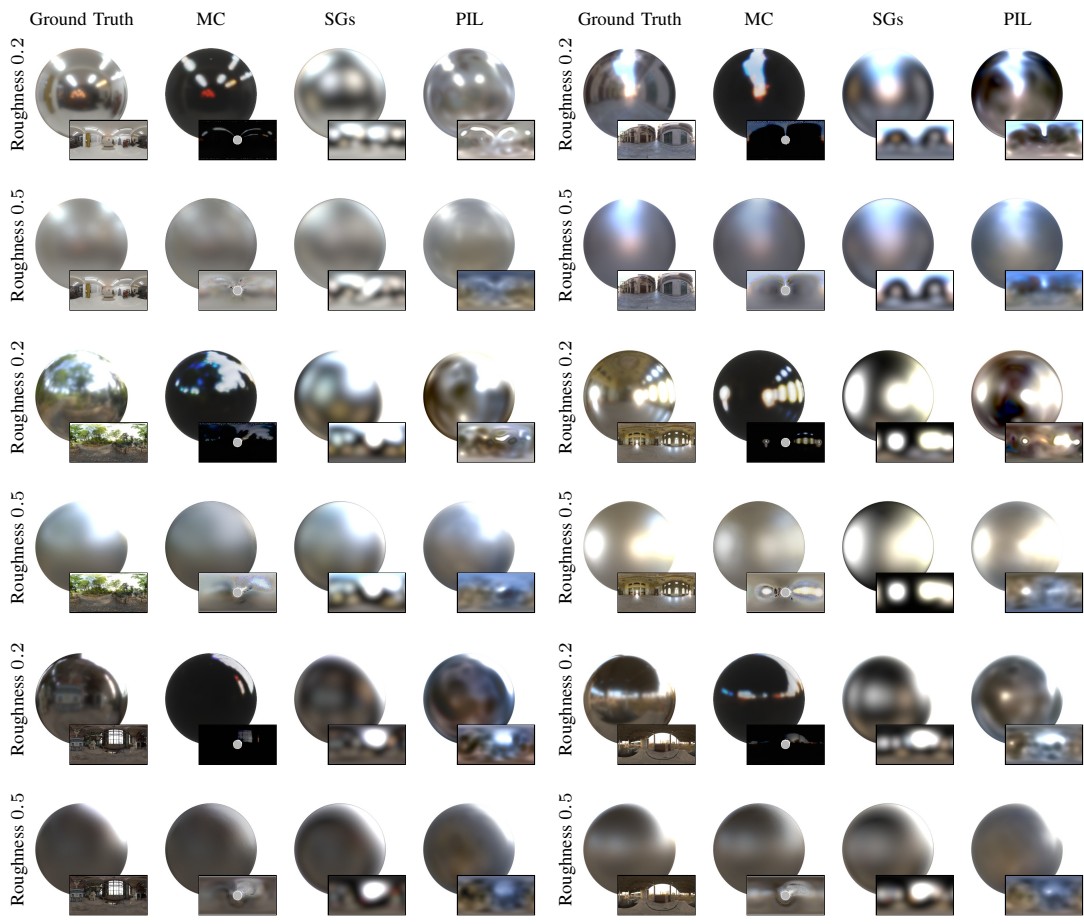

Figure A1: **Neural-PIL vs. SGs.** Optimized illuinations with SGs and our Neural-PIL.

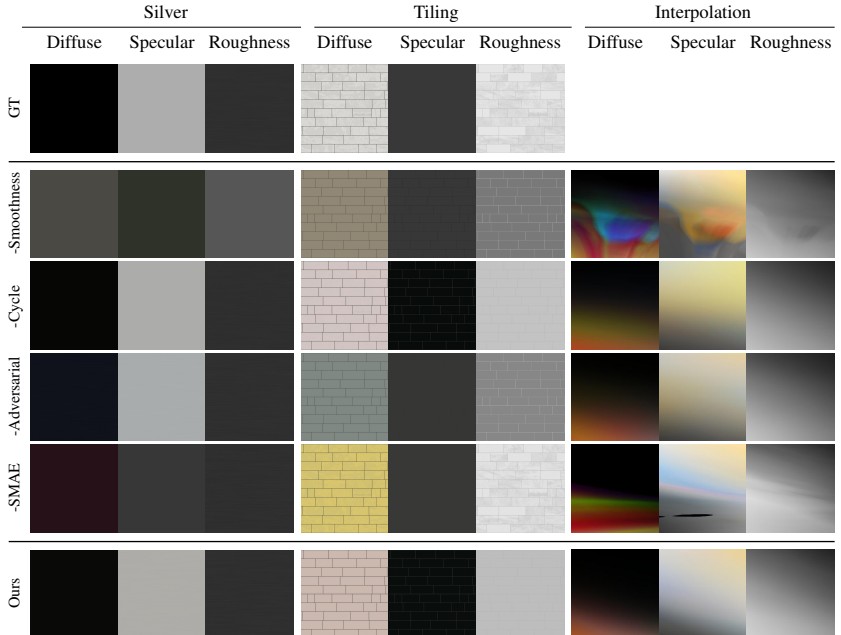

Figure A2: **BRDF SMAE Ablation.** Sample BRDF optimizations and interpolated BRDF space with SMAEs learned with the exclusion of different loss functions.

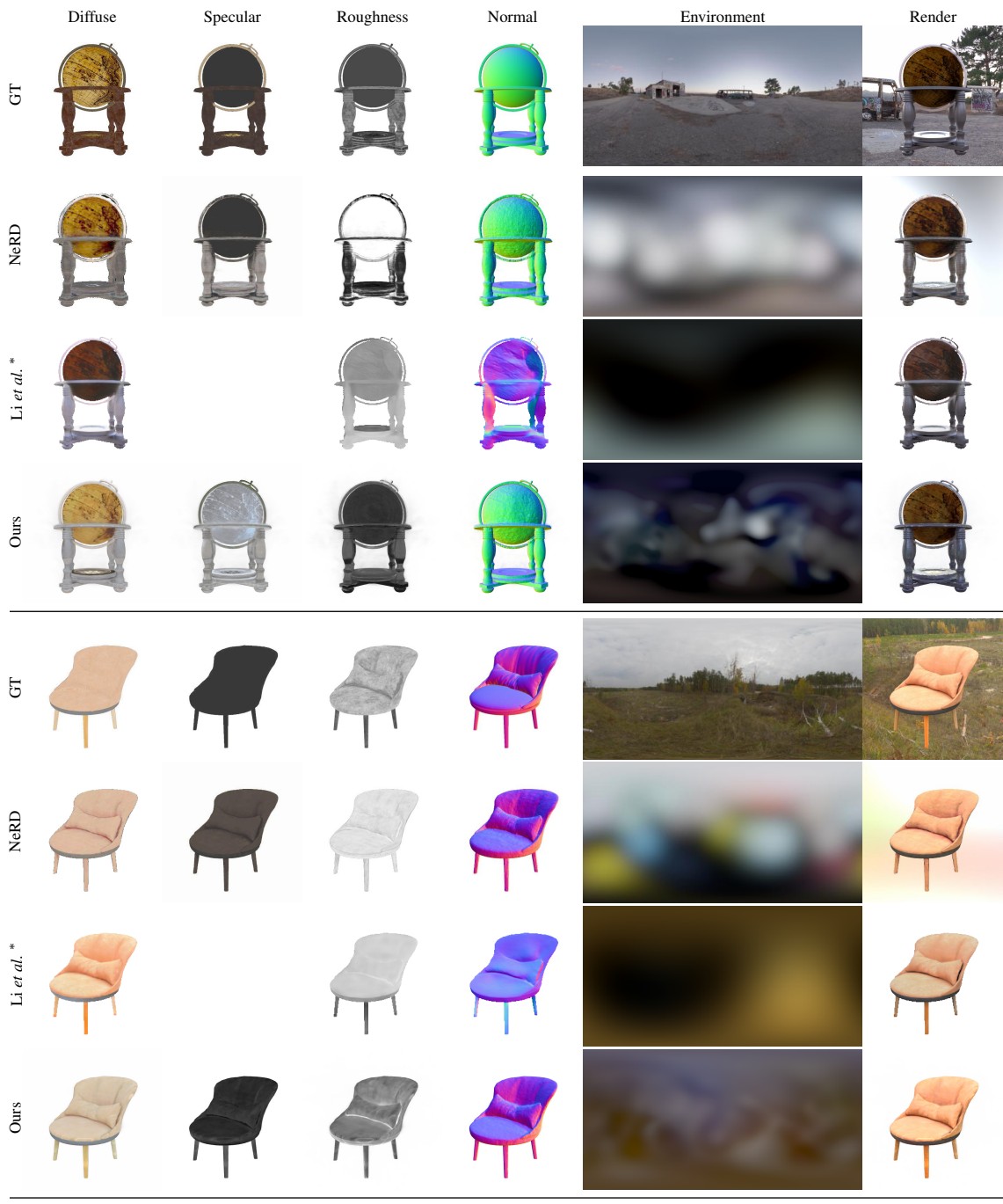

Figure A3: **Additional BRDF Decomposition Results.** Comparison with NeRD [2] and Li *et al*. [5] on two synthetic scenes.

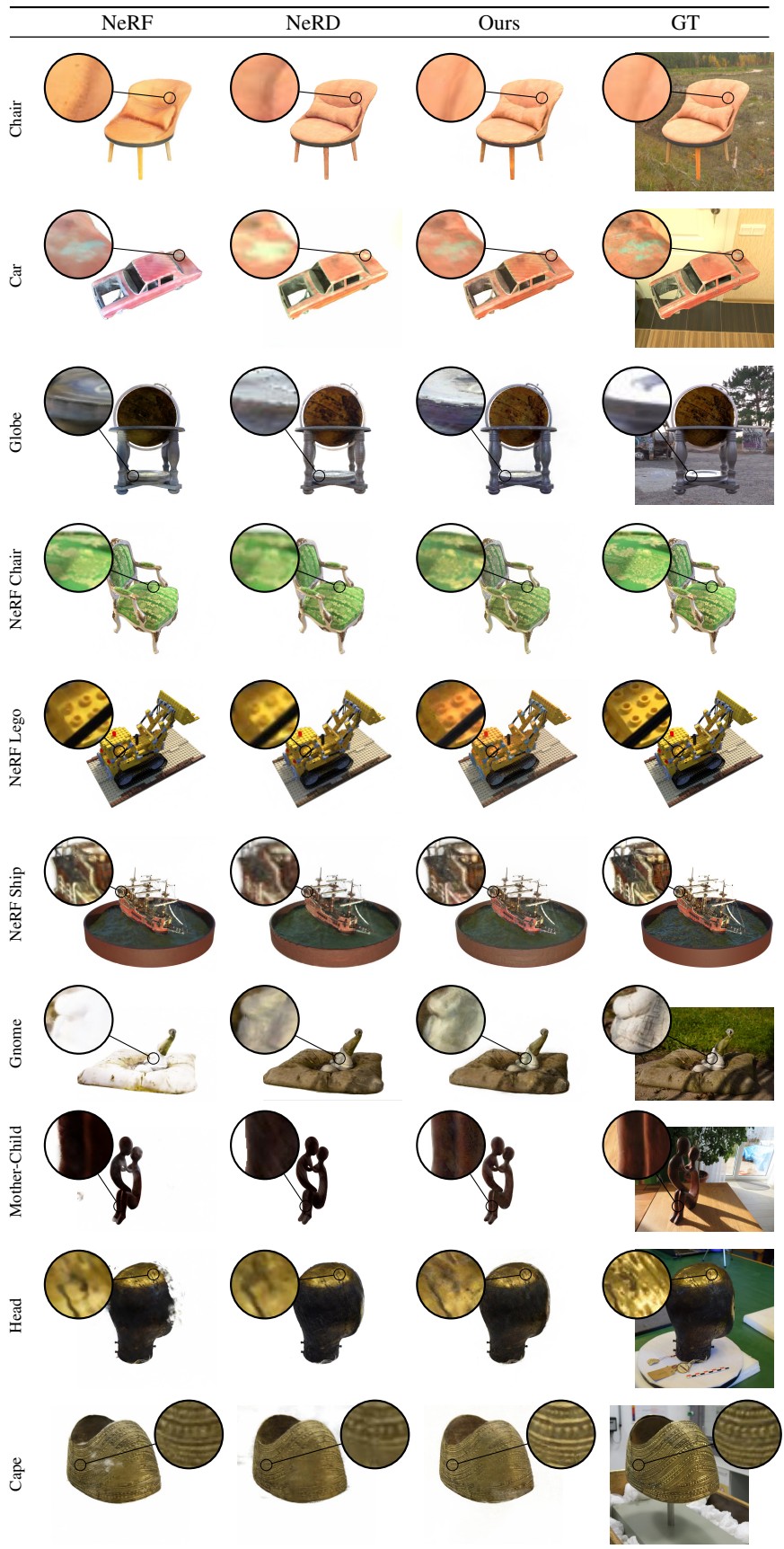

Figure A4: **Results from each scene.** Comparison with NeRF, NeRD and Neural-PIL for every scene.

| | Base Image | Illumination 1 | Illumination 2 | Illumination 3 |
|---|---|---|---|---|
| Car | | | | |
| Chair | | | | |
| Gnome | | | | |
| Mother-Child | | | | |
| Cape (Rotate) | | | | |

Figure A5: **Additional relighting results.** Relighting of various scenes under the source illumination shown in the insets. For the last row the illumination is rotated.

[6] Rosanne Liu, Joel Lehman, Piero Molino, Felipe Petroski Such, Eric Frank, Alex Sergeev, and Jason Yosinski. An intriguing failing of convolutional neural networks and the coordconv solution. In *Advances in Neural Information Processing Systems (NeurIPS)*, 2018.