# OpenReview forum: "Neural-PIL: Neural Pre-Integrated Lighting for Reflectance Decomposition"
_NeurIPS.cc/2021/Conference — NeurIPS 2021 Poster_

### Official Review · Reviewer_6Uz6 · 2021-07-16

**Rating:** 6
**Confidence:** 5

**Summary:**

The method suggests to learn a 3D model of shape, reflectance and light from a set of 2D images. They use a forward model that combines the three factors and invert it by optimizing its parameters to explain the 2D images. The key is in the specific parametrization of the three factors.

The first factor is a classic implicit model of shape. Rays are cast from the views of the 2D input images to find 3D depth. They define it as expected opacity-weighted depth, which might have its problem on (soft) silhouette edges. Mode finding, via Iterated Re-weighted least squares would be still fast and maybe more correct, not resulting in depth values floating in the air.  This said, this part is common.

The reflectance is parametrized as a spatial field of 4D latent codes. Each latent code itself is an embedding of the 7D Cook-Torrence parameters over the MERL database. The paper says the 7D codes are "not constrained enough", but no evidence for this is given. The embedding is pre-trained using a modern smooth auto-encoder.

The illumination is parametrized in a similar way: a smooth auto-encoding of a collection of environment maps into a low-dimensional code.

The work dwells a bit too much on how this is a graat AE'ing, but it is just shopped previous work, so it should maybe be phrased more neutral.

Now the key innovation is how the two latter factors combine. The direct way is to solve an integral where the BRDF slice is convolved with the illumination. in SH this is a dot product, but at the expense of angular details. Instead, the suggestion is to pick up the realtime-rendering idea of pre-filtered illumination. This allows to get the integral estimate in constant time, after pre-processing the envmap. Now instead of preprocessing the envmap (which would be possible, there is only one illumination per scene and pre-process = pyramidal blur = constant time), authors suggest to use a network that directly returns pre-filtered results. For this idea to be fully appreciated, one would need to look at the two ablations chosen by those familiar with game graphics: pixel basis (MIP maps) and MC directly. This would need to be compared against an NN doing this pre-filtering job. It is a cool idea, but does it do anything in the end? Comparison is only made against SH, which is for the birds to those learned in graphics. The pixel basis ablation is: take the envmap code, decode into pixel basis, MIP, use it as Karris explains with the 7D shading model parameters. The MC baseline is, take envmap code, decode into pixel basis, MC it with the 7D shading model parameters as in any path tracer (eventually, importance-sample). Clearly MC noise hurts human eyes, but does it also hurt SGD convergence? We remember Noise2Noise and other instances were training on unbiased noisy data in the grand scheme of SGD make no difference.

Instead of those ablations, focused on what is proposed, the paper compares to an Arxiv paper and a published alternative by Li et al. Comparison to Li et al is okay, we need to use something, done. Comparison to NeRD is more questionable. NeRD is very similar to this work. It more appears this paper is a fix to NeRD than a true baseline. Most agree, Arxiv papers cannot be held AGAINST a submission. Now authors might agree that not every success over an Arxiv paper indicates something. Otherwise, Arxiv turns into a "strawman repository" and everything is even more confusing then it already is these days.

For results, one would now expect to see something specular. As Fig. 6 shows, this is the particular strength of this approach. Alas, there was none. Results are very small, they are the same data s in NeRD. For relighting, this reviewer is very unsure what to compare this to? The Graphics approach is usually to make a photo under the novel light condition and then see how much the AI could predict it. Here we only see an image.

**Ethical Concerns:**

No idea

**Ethics Review Area:**

["I don’t know"]

**Limitations And Societal Impact:**

No idea

**Main Review:**

See above. This reviewer preferred to merge description of the work and critique in one field in this case.

This reviewer is unsure about what to recommend. As an *increment* over NeRD, it will not pass. It does not analyze the benefits of pre-integration nor does it show results (shiny reflections) where this really contributes. As a *replacement* for NeRD, this is a unique system that deserves publication. There is concurrent work this method appears to be promising to outperform, it is simpler to code and pre-integration not yet understood enough in Computer Vision probably. Authors would need to clarify what is the position, replacement or increment?

If suggestion is a replacement then the writing would need to change and instead of shooting down a NeRD baseline from the "straw man repo Arxiv", the text needs to be reworked and NeRD has to be "dissolved" into a "strong ablation".

**Time Spent Reviewing:**

3 h

---

> ### Author Response · Authors · 2021-08-10
> **Response to 6Uz6**
>
> **On the BRDF encoding**: Our dataset is not derived from the MERL database but from Boss et al. CVPR’19 - Two-shot SVBRDF and Shape estimation, which directly uses parameter maps for the Cook-Torrance model. In NeRD the authors found that directly trying to predict diffuse and specular albedo colors to be challenging, as these values are inherently linked and need to conform to specific rules to be physically plausible. The authors proposed to use a basecolor-metallic material model as it provides this constraint. As this constraint was helpful, we wanted to evaluate a learned manifold of materials that enables the full diffuse and specular parametrization as it enables higher flexibility and can express more edge case materials.
> On the novelty of the SMAE space: We introduced several novel losses which only enable this smooth manifold generation. Without these losses, the previous methods do not generate this nice manifold which is essential in a generative latent code optimization (GLO).
>
> ***
>
> **On pre-processing the envmap**: Yes one can easily blur an envmap and precompute the integral as done in real-time rendering. A simple blur is only sufficient for the diffuse component of the BRDF. The specular component needs a complex blur which follows the GGX distribution function and needs to be computed for several roughness levels and blending between the discrete steps, as our materials are spatially varying and contain different roughness values. This step is not cheap as many samples need to be integrated to provide a smooth and noise-free integration. One could employ dual importance sampling where first the importance of the environment map is computed and then the product with the GGX importance sampling is calculated to reduce the sample count. This step is also not cheap to compute and needs to be recomputed at every training step, as we are learning the environment map itself. Additionally, our method is capable of dealing with varying unknown illuminations per scene, where each image is captured under different illumination. Computing these integrations during training is therefore infeasible and led us to develop Neural-PIL.
>
> ***
>
> **On directly using MC and dealing with the noise**: The main issue is that the envmap is not known. So importance sampling is only possible on the BRDF or it needs to be recomputed after every step. Rendering research has shown that sampling either of them instead of their product produces noisy results. The difference to Noise2Noise is that we are not optimizing an image but the underlying BRDF. For example, if a bright spot in the image is our target and we miss a small light source, the optimization has to attribute it to the texture. The texture becomes brighter. Now if the same spot is optimized again but now from a different image, the optimization needs to correct the texture change. Dealing with these conflicting signals would at least slow convergence drastically or in the worst case generate a local minimum.
>
> ***
>
> **On visual comparison**: See the above general response: **More visual comparisons on real-world scenes**. Only the Real-world and synthetic data with varying illumination have varying illumination in the test sets.

---

> > ### Comment · Reviewer_6Uz6 · 2021-08-16
> > **Too subtle for those results**
> >
> > Agreed, the BRDF parameters are not from MERL but from a different database. Not sure this maters. These materials shown are far from complex, far from subtle. Instead they are just-a-bit-glossy, at best. This reviewer remains unconvinced that for this result quality, uniform random covering of Phong parameter space is not maybe also enough.
> >
> > Other reviewers also note that a for loop over envmap pixels (quadrature) might do the same. The rebuttal correctly explains what are the drawbacks of pre-filtering (it is hard to get GGX correct). Also I have already acknowledged the drawbacks of MC (noise) and authors correctly say importance sampling is further challenges. If these problems were real that would matter; it would be arguments for PIL. But while all that is said is correct (GGX incorrect, MC noise), it is not clear how much such things matter for this level of material reproduction and learning. If all that happens is to make a small view-dependent bump, that will work in GGX as it will work in Phong or the most naive basis, noisy training images or not. A for loop or MC also is not difficult a baseline to code or explain, either.
> >
> > In summary, the result quality (and quantity) does not hold up to the subtle (mostly correct)  arguments why those technical choices are required.

---

> > > ### Author Response · Authors · 2021-08-18
> > > **On using MC as PIL replacement**
> > >
> > > Following your suggestion, We have implemented a baseline that uses differentiable MC integration with BRDF importance sampling. Currently, we take 128 samples per pixel (spp) and optimize a 128x256 resolution environment map. A lower resolution environment map does not lead to better results but the pixelization is clearly visible. With fewer spp, the optimization becomes unstable using gradient descent and the quality decreases especially for the non glossy material.
> > >
> > > We have added the MC baseline to our PIL/SGs comparison sphere images. Here we use a rendered image of a sphere in a resolution of 400x400 as the target and we know exactly the geometry and the BRDF. The goal is then to use the corresponding method to fit the illumination so that the target is replicated as close as possible. All pixels are optimized in a single batch. The following table shows the memory and runtime comparisons for optimizing different types of illuminations:
> > >
> > > | Method | GPU Memory | Time for 1000 optimization steps |
> > > |---|---|---|
> > > | MC 128spp | 18GB | 6 minutes |
> > > | 24 SGs |  4GB   |  1 minute  |
> > > | PIL (Ours) | 8GB | 2.5 minutes |
> > >
> > > This runtime and memory overhead with MC integration is quite high making it infeasible to use inside our decomposition framework. Making the differentiable MC integration with importance sampling fast and efficient forms an interesting future research direction.
> > > We have added the MC optimized illuminations and renderings to the supplementary figure:  [Anonymous Link](https://www.dropbox.com/s/d0s6b29v9qp33ta/NeuralPIL-Supplement-MCRendering.pdf?dl=0). With the addition of MC-based rendering, Table R1 with comparisons across different illuminations now becomes:
> > >
> > > **Table-R1**: *Average PSNR on rendered spheres with MC, SGs and Neural-PIL illuminations.*:
> > >
> > > | Roughness Value | MC (higher) |SGs (higher) | Neural-PIL (higher) |
> > > |---|---|---|---|
> > > | 0.2 | 34.88 | 31.57 | **35.76** |
> > > | 0.5 | 35.14 | 28.98 | **35.28** |

---

### Official Review · Reviewer_rZtL · 2021-07-17

**Rating:** 6
**Confidence:** 4

**Summary:**

This paper proposes to learn a neural implicit representation from multiple images under different illumination that can separate complex SVBRDFs and lighting, which can be used for applications such as view synthesis and relighting. There are two major contributions: one is two learned manifolds to constrain BRDF and lighting parameters that can better preserve high-frequency information, the other is an MLP based network called Neural-PIL that can render intensity through a single forward pass based on the new lighting and BRDF representation. Experiments show that the learned lighting manifold can better recover details of the environment map compared to traditional spherical Gaussian representation, and the proposed method can achieve more accurate view synthesis and relighting results compared to the baseline NeRD.

**Ethical Concerns:**

I cannot see any ethical concerns.

**Limitations And Societal Impact:**

Authors have listed the limitations quite completely in section 5. Maybe one more limitation to add is that the proposed method may completely rely on the geometry prediction from the coarse network and cannot optimize the geometry jointly with BRDF and lighting. The reason is that the current method only computes color at the surface points.

I cannot see any potential negative societal impact.

**Main Review:**

I will list the strengths of the paper and my questions. When listing my questions, I will separate them into major questions that may change my evaluation and minor questions. Authors can focus on major questions in the rebuttal.

Strengths:
1. The idea of using learned manifolds to constrain lighting and BRDF reconstruction is interesting. Given the ill-posed nature of inverse rendering under an unconstrained environment, such constraints are necessary to achieve satisfactory reconstruction results. Besides, the new lighting manifold is shown to better preserve high-frequency lighting compared to spherical Gaussian representation but with much fewer parameters. This new representation may inspire future research too.

2. It is great to use a pre-filtered environment map to reduce the rendering cost. While this method is popular in the graphic community, it is the first time I see people use that to help to solve inverse rendering problem.

Major questions:

1. The writing of the paper may need to be improved. Many important details are probably missing.

a. I have a hard time understanding how the Neural-IPL is trained. Line 250 and 251 do not mention explicitly that Neural-IPL is trained first and fixed through the rendering process. Only when I check the supplementary material, did I understand this part. Authors can consider moving more training details from the supplementary material to the main paper to make the main paper easier to follow.

b. Line 182: It is unclear to me how the termination point is computed. I guess it is probably done by computing the weighted average based on the transmittance but more details should be included in the paper.

c. Table A1 in the supplementary only summarizes the network structure of Light-SMAE. It will be good to have similar summaries for other networks.

d. Line 192: It has been established before that we will only use $w_r$ as an input to render the color, which causes it to be very confusing to say Neural-IPL takes $w_i$ as input here. This is explained later in Line 197 but the whole paragraph may still be confusing to read. We can just say the input is $w_r$. In addition, the $w_i$ in Figure 3(b) can be changed to $w_r$.

e. It may not be explained in the paper if the illumination of the input image changes, how we should handle lighting code $z_l$. Do we always have different lighting codes for every input image or that is only true when the illumination changes?

2. The experimental results can probably be more convincing.

a. There is only 1 real example shown in the paper. And for this real example, it is difficult to observe specular highlights. There is no reconstructed environment map for real examples shown in the paper.

b.  Table b shows that the proposed method achieves more realistic view synthesis results compared to NeRF on their synthetic dataset (Ship, Chair, Lego). Line 312 mentions that it is because the proposed method has more realistic view-dependent effects. It will be very interesting to show these examples in the paper, especially because the reconstructions of NeRF are already quite impressive. This can be very compelling results to show the advantages.

Other issues:

1. Line 188 to 191 says that Monte Carlo sampling or numerical accumulation is not practical. But if the environment map resolution is only 128 x 256, the computational cost of directly computing the integral is probably lower than the MLP. The main argument for Neural-IPL is probably that $z^l$ provides a good prior, not the computational cost.

2. It will be interesting to show a comparison of view synthesis and relighting results with and without the BRDF embedding. Since BRDF embedding only reduces the dimension from 7 to 4, I am curious how significantly the results can be improved.

3.  The normal map in Figure A3 of the supplementary material is wrong. It probably shows the normal map in the world coordinate system instead of in the camera coordinate system.

Minor questions:

1. Line 290: are shows in Figure 6 -> are shown.

While I like the idea of using learned manifold to constrain lighting and BRDF reconstruction, especially the lighting reconstruction, I think the current experiments and writing may have space for improvement. Therefore, I keep my review to be slightly negative. I would like to see  authors' rebuttal and update my review.


After rebuttal: authors rebuttal solved my questions and I upgraded my review to be positive. Please refer to the rebuttal discussion for details. Thanks authors for addressing my questions.

**Time Spent Reviewing:**

5

---

> ### Author Response · Authors · 2021-08-10
> **Response to rZtL**
>
> **On moving training details to the main paper**: Thanks for the suggestion. We will revise the main section on the training details to clarify the pipeline.
>
> ***
>
> **On the computation of the termination point**: The termination is computed based on the opacity along the ray as described in line 129. Solving this integral for $T(t)$ using numerical quadrature gives the expected depth. We will reference this in the final version.
>
> ***
>
> **Summaries of network structures**: Good point. We will include the details of the remaining network structures.
>
> ***
>
> **On $\omega_r$ and $\omega_i$ notation**: Thank you for the catch. We will fix this.
>
> ***
>
> **On handling the illumination codes $z_l$**: Currently, we handle two scenarios: either each image has its own illumination or all images have shared illumination. However one could easily extend it so that some images share an embedding.
>
> ***
>
> **On missing visual results**: See the above general response: **More visual comparisons on real-world scenes**
>
> ***
>
> **On direct MC sampling of environment maps**: Even low-resolution maps would require an interpolated sampling, as for glossy materials a specific sample direction is required. If we only sample a low-resolution map, the actual direction might lie between the points. For perfectly Lambertian materials one can indeed sample all pixels belonging to the hemisphere and this might be faster, but when the glossiness (specularity) is increased this would require an interpolated sampling with a large number of samples to reduce noise. This is then significantly more expensive.
>
> ***
>
> **On BRDF-SMAE influence**: See the above general response: **Ablation study of BRDF-SMAE**
>
> ***
>
> **On world-space normal map visualizations**: Thank you for the suggestion. We will convert the normal maps visualization to camera coordinates.

---

> > ### Comment · Reviewer_rZtL · 2021-08-22
> > **Further questions**
> >
> > Authors solve many of my questions. But I may have two more questions.
> >
> > 1. On direct MC sampling of environment maps: I feel 128 x 256 is actually a very high-resolution environment map. If you do importance sampling for a glossy surface with 128 x256 samples, I am almost sure the results will be very good. Meanwhile, if you have a 512 layers MLP, the computational cost is probably already higher.
> >
> > 2. On new results show in the PDF: Some new results look weird, for the Lego result, why the new method produces results with a different color. For the Genome result, why NeRF result has a very different color compared to the ground-truth?

---

> > > ### Author Response · Authors · 2021-08-24
> > > **On MC sampling and experimental results**
> > >
> > > **Question 1**: Reviewer 6Uz6 also suggested using MC sampling of environment maps and we have prepared an experiment using differentiable MC with importance sampling of the BRDF. We take 128 samples per pixel (spp) based on the BRDF importance sampling and learn a 128x256 environment map. Overall we found the memory overhead to be quite severe with 128 spp requiring over 18GB on our sphere images (see our answer **On using MC as PIL replacement** for Reviewer 6Uz6). Lowering the spp improves the memory consumption in this regard but the downside is a noisy, unstable gradient for rough materials, as the few samples cover nearly the entire hemisphere. As an unstable gradient prevents an accurate optimization and the memory requirement is quite high compared to NeuralPIL it would be challenging to include MC sampling in a full decomposition framework. Overall the results provide well-detailed highlights ([Anonymous Link](https://www.dropbox.com/s/d0s6b29v9qp33ta/NeuralPIL-Supplement-MCRendering.pdf?dl=0)) but all areas except the highlights are not captured well.
> > >
> > > We agree that MC sampling is a good forward model for high-quality rendering. But, what we are interested in is the *backward process* that estimates illumination, shape, and material properties by *back-propagating* through the rendering process. We are not aware of any works that successfully used MC sampling for such a complex optimization problem.
> > >
> > > **Question 2**: Good observation on the color shifts in the Lego scene. Our hypothesis for this behavior is due to the artificial illumination used in this scene. The Lego set is placed in complete darkness and only illuminated by two large area lights. As our NeuralPIL is trained on natural illuminations and we enforce staying on the learned manifold, to ease the optimization, the illumination is reconstructed as the closest fit on the manifold of natural illuminations. Our BRDF-SMAE is also constrained on a manifold of natural BRDFs and with both constraints, this artificial setup could not be perfectly recovered. The Gnome scene features varying illumination, which NeRF is not capable of representing, and showcases the need to handle relighting, which our method enables - even under completely unseen illuminations.

---

> > > > ### Comment · Reviewer_rZtL · 2021-08-24
> > > > **Questions about the qualitative results**
> > > >
> > > > Thanks authors for the explanation on MC sampling experiment. I am quite surprised that MC sampling has such large GPU memory consumption, which is interesting to know.
> > > >
> > > > I am still having a hard time to interpret the view synthesis and relighting result. In table 1 (b), it shows that the proposed method achieves better SSIM on single lighting view synthesis on synthetic data. But looking at the result, especially given the color shift on some examples, I cannot see why the new results are better than NeRF. In table 1 (c), authors show the results on real world view synthesis and relighting, but I cannot find real-world relighting results. Sorry if I miss something, but can authors point me to the results. Thanks!

---

> > > > > ### Author Response · Authors · 2021-08-30
> > > > > **On our results**
> > > > >
> > > > > **On NeRF vs. Our results for single lighting view synthesis**: The following table shows different metric comparisons between NeRF and Our results on the 3 synthetic scenes with fixed illumination.
> > > > >
> > > > > **Table-R4**: NeRF and Neural PIL comparison on the NeRF dataset synthetic data
> > > > >
> > > > > | Type | Scene | NeRF-PSNR (Higher) | NeRF-SSIM (Higher) | NeRF-LPIPS (Lower) | Ours-PSNR (Higher) | Ours-SSIM (Higher) | Ours-LPIPS (Lower) |
> > > > > |---|---|---|---|---|---|---|---|
> > > > > | Syn. Fixed | Chair | 30.5258 | 0.96 | 0.0664 | **31.4396** | **0.9801** | **0.0336** |
> > > > > | Syn. Fixed | Lego | **31.0264** | **0.9709** | **0.0268** | 27.8887 | 0.9642 | 0.0426 |
> > > > > | Syn. Fixed | Ship | **28.0942** | 0.9168 | 0.0876 | 26.4081 | **0.9180** | **0.0707** |
> > > > >
> > > > > As the results indicate, the metrics also capture the dip in our result quality compared to NeRF in the Lego scene. Ours-SSIM is better in the other two scenes. In general, we have noticed finer details in the results compared to NeRF. As SSIM is more sensitive to structural details we achieve a higher score compared to NeRF in this metric. Reasons for the improved details might be attributed to differentiably optimizing the normal that is derived from the density field. So the photometric loss per ray can also improve the geometry in a neighborhood instead of only at that specific point. In addition, the shape-radiance ambiguity in NeRF might be also reduced due to our explicit rendering step.
> > > > >
> > > > > **On real-world relighting results**: In Fig. 7b, we show a relighting result with a random different illumination and we show the illumination transfer result in Fig.1.  Additionally, we have generated another figure ([Anonymous Link](https://www.dropbox.com/s/leqvwn3pyubhbqs/NeuralPIL-Supplement-RWRelightAddition.pdf?dl=0)) where we transfer illumination between test images with the target illumination estimated from the images in insets. For the cape, we just show a rotating environment illumination.

---

> > > > > > ### Comment · Reviewer_rZtL · 2021-08-30
> > > > > > **Thanks authors for the new results**
> > > > > >
> > > > > > Thanks authors for the new result! Given the new results I would like to change my rating to be positive. Authors rebuttal has solved most of my questions in the rebuttal. If possible, I still hope that authors can show more real example results in the final version. I especially like the idea of using a manifold to constrain lighting estimation.

---

> > > > > > > ### Author Response · Authors · 2021-09-01
> > > > > > > **Thanks for the positive response**
> > > > > > >
> > > > > > > Glad to see your positive response. Looking forward to your score update.

---

### Official Review · Reviewer_pHT7 · 2021-07-18

**Rating:** 6
**Confidence:** 5

**Summary:**

The paper presents a method to reconstruct geometry and BRDF from a set
of multi-view images take under various illuminations. It learns a low
dimensional manifold for illuminations and BRDFs and use these data priors
to regularize the estimation. It also applies a network to predict the
integral of light for efficient rendering under environment maps. It
shows results on both synthetic and real scenes.


**Limitations And Societal Impact:**

Please see main review.

**Main Review:**


### Strengths:
1. Using networks to predict the integral of lighting is novel and
reduces the computation cost for rendering under complex lighting conditions.

2. Using auto-encoders to learn a data manifold for BRDFs and lighting is interesting
and helps regularize the prediction.

### Weakness:

1. The results seem to be incomplete. The paper says it has results on
4 real scenes, however, only results on the cape scene are showed.
Moreover, for the cape scene, only several images from different views are shown.
It is desired to show more video results on more diverse scenes under different illumination conditions
and viewpoints, and include comparisons to baselines. Similarly
I cannot find the view synthesis results on Ship, Chair, and Lego scenes.
From the current results, I cannot judge whether the proposed method can
generate convincing results on real scenes.

2. In terms of comparison to NeRF and NeRD in Figure 7, it is confusing to me
why the proposed method would outperform NeRF even for view synthesis tasks.
In fact from the video, I think the results of NeRF has more details while the
proposed method generates blurry renderings. It is desired to show quantitative
and qualitative results on the NeRF dataset to further validate this claim.

3. In the novel view and relighting results in the video, ground truth should
be included. The current results are not fully convincing, for example, there
are clear artifacts on the globe scene.

4. It is not clear to me whether the proposed method requires images under
varying illuminations. Requiring images under different environment
lightings will make the acquisition very difficult and make the method impractical.
How does the performance change with images under fixed and varying illuminations?

5. In terms the comparison on lighting, while the paper claims Neural-PIL has
better performance, from the results in Figure 6 (second row) and the results in
supp (Figure A1), I think the Neural-PIL illuminations often suffer from incorrect
colors and highlight positions. Therefore it's not convincing to me that Neural-PIL
is definitely better. To better show this, a quantitative comparison should be included.

6. The paper uses an auto-encoder to regularize the material estimation. While the paper
does ablation on different loss terms in training the auto-encoder, a more important
and intuitive comparison is how does the full pipeline work with and without using
the auto-encoder? How does the method work by directly predicting the BRDF coefficients
without the auto-encoder?

7. Another important component to validate is the accuracy of the PIL network. It is not
certain to me that such a network can be trained effectively without direct supervision.
A comparison between the predicted integrated lighting and the GT integral on synthetic
data would be helpful.

Overall while I like some components and design of the paper, I think the paper needs
significant improvement in terms of visual quality and experiments to further validate the
effectiveness of the method.

**Time Spent Reviewing:**

2

---

> ### Author Response · Authors · 2021-08-10
> **Response to pHT7**
>
> **On missing visual results**: See the above general response: **More visual comparisons on real-world scenes**
>
> ***
>
> **Performance improvement compared to NeRF**: Visually NeRF can generate sharper and more detailed results. However, NeRF has the ability to model transparent effects, which introduces another degree of freedom to express the images. Highlights are often added as floaters and therefore hurt the numerical evaluations. Our physical constraints aid in reducing floaters but at the cost of losing transparency effects.
>
> ***
>
> **On performance under varying and fixed illumination**: Our method tackles both capturing scenarios and we present results for both the settings. While varying illumination proves more challenging in regards to illumination estimation, the BRDF and illumination separation is better constrained. For a fixed illumination a more precise illumination can be estimated, but the capture setup is restricted to in-studio image capturing or a rapid thorough capture where no recapture for missed angles is possible, as the illumination will be different across recaptures. Additionally, ambiguities in separating BRDF and illumination are harder to disentangle, as e.g., darker areas can be attributed to either the texture or a shadow from the illumination. We favor the higher degree of practicality from varying illuminations, as this allows us to leverage online image collections which are mostly taken under different illuminations.
>
> ***
>
> **On Neural-PIL evaluation**: See the above general response: **Neural-PIL vs SGs quantitative evaluation**
>
> **On directly predicting the BRDF without BRDF-SMAE**: See the above general response - **Ablation study of BRDF-SMAE**

---

> > ### Author Response · Authors · 2021-09-01
> > **Concerns w.r.t. results and comparisons**
> >
> > Thanks again for your constructive feedback. We hope our responses have addressed your concerns w.r.t. results and comparisons. Please let us know if you have further concerns or suggestions. Thanks

---

> > > ### Comment · Reviewer_pHT7 · 2021-09-02
> > > **Reviewer feedback**
> > >
> > > I thank the authors for the rebuttal. The replies and additional results address my concerns. I increased my scores to 6.

---

### Official Review · Reviewer_rECe · 2021-07-26

**Rating:** 6
**Confidence:** 4

**Summary:**

This paper proposes a method for joint shape, BRDF and illumination estimation using a NeRF-like pipeline. Compared with NeRF, the proposed method can also solve for illuminations, and it  outperforms NeRD on both quality and efficiency due to a novel network query-based pre-integrated lighting approach.

**Limitations And Societal Impact:**

See Weaknesses above.

**Main Review:**

### Strengths:

1. The idea of network query-based pre-integrated lighting is novel, and combining it with smooth manifold auto-encoder (SMAE) in a differentiable  pipeline is non-trivial.

2. The proposed Neural-PIL achieves the state-of-the-art in both quality and efficiency.


### Weaknesses:

1. The flowchart in Fig. 3 can be improved, e.g., I did not see Light-SMAE in this figure. Moreover, the training strategy in L251-257 seems complex, e.g., why freeze BRDF-SMAE’s decoder only, any experimental support? It would be good if the authors can elaborate more on this in rebuttal.

2. The SMAE ablation study (in supplementary) provides nice visual comparisons, however it lacks quantitative comparisons. In Fig. A1, for example, although Neural-PIL shows finer details than SGs, its color looks blueish (also in Fig. 6), therefore it's hard to tell whether the PSNR/SSIM are better. I'd like to hear some explanations on this in rebuttal.


**Time Spent Reviewing:**

4

---

> ### Author Response · Authors · 2021-08-10
> **Response to rECe**
>
> **Light-SMAE is missing in Fig-3 and the training strategy seems complex**: The Light-SMAE is not directly used in neural volume training. It is only used for pre-training to learn the manifold for the Neural-PIL network. So the steps before training the neural volume are Light-SMAE -> Neural-PIL -> Neural Volume training. We’ve split the Neural-PIL training for efficiency reasons as training both SMAE and PIL networks at the same time limited the batch size drastically. The Neural-PIL and the BRDF-SMAE are then included in a frozen manner and only the embedding is allowed to be optimized to keep the learned manifold intact. If the network is not frozen, the carefully crafted manifold is destroyed and the regularisation properties of both networks are nullified. We will add this discussion on the overall training process.
>
> ***
>
> **On the blue tint in the PIL evaluation**: See the above general response - **Neural-PIL vs SGs quantitative evaluation.**
>
> ***
>
> **Missing BRDF-SMAE ablation study**: See the above general response - **Ablation study of BRDF-SMAE**

---

> > ### Comment · Reviewer_rECe · 2021-09-02
> > **Final rating**
> >
> > Thank you for the detailed response and new experiments, they addressed most of my concerns, and I'll keep my original rating.

---

### Author Response · Authors · 2021-08-10
**General Response**

We thank all the reviewers for their constructive feedback. We are glad that reviewers appreciated the novelty of the neural pre-integrated illumination network (Neural-PIL) as well as the use of smooth manifold auto-encoders (SMAE).
Common concerns among the reviewers include missing ablations of Neural-PIL and SMAEs; as well as only few visual results on real-world scenes. We first address these common concerns here, followed by responses to each reviewer’s concerns.

***

**Neural-PIL vs SGs quantitative evaluation**: We computed the average PSNR on the rendered spheres of all the examples shown in the supplementary figure A1 w.r.t GT renderings. Table-R1 shows the mean PSNR of these 6 scenes:

**Table-R1**: *Average PSNR on rendered spheres with Neural-PIL and SGs.*:

| Roughness Value | SGs (higher) | Neural-PIL (higher) |
|---|---|---|
| 0.2 | 31.57 | **35.76** |
| 0.5 | 28.98 | **35.28** |

Overall Neural-PIL (Ours) achieves a 4-7 increase in PSNR, which is a considerable improvement. We want to re-emphasize that the rendering time with Neural-PIL is also two orders of magnitude faster.

*Reasoning for bluish tints*: Neural-PIL optimization starts with a mean embedding of our HDRI dataset which is gathered from an online collection with most of them being outdoor illuminations with blue sky. For roughness of 0.5, Neural-PIL needs to integrate a large portion of the environment map and light behaving additive, the optimization just counteracted the blue sky with a slightly more yellowish tint in other parts. Despite having bluish and yellowish tints in the decoded environment map, the resulting renderings with Neural-PIL more closely resemble the GT compared to using SGs (both quantitatively and qualitatively). For lower roughness of 0.2, Neural-PIL needs to integrate light locally and thus maintain more local information of the environment map (as shown in supplementary figure A1). We will clarify this in the paper.

***

**Ablation study of BRDF-SMAE**: The following Table-R2 shows BRDF estimation error on synthetic scenes using our full model with and without BRDF-SMAE.

**Table-R2**: *Average error on estimated BRDF parameters with and without using BRDF-SMAE*:

| Method [MSE] | Diffuse (Lower) | Specular (Lower) | Roughness (Lower) |
|---|---|---|---|
| w/o BRDF-SMAE | 0.0649 | 0.1192 | 0.0233 |
| Full Model | **0.0095** | **0.0207** | **0.0033** |

Results clearly show that  BRDF estimation suffers drastically without BRDF-SMAE.

***

**Ablation study of Light-SMAE**: Light-SMAE is not a part of our final reflectance decomposition network, but is only indirectly used via pre-training for the Neural-PIL network. Without Neural-PIL, our decomposition network would be very similar to NeRD which we compare with in the paper.

***

**More visual comparisons on real-world scenes**: We deeply regret missing including further real-world visual comparisons. This is a mistake on our side and we do not want to make the impression that the results in these scenes do not work. We have prepared a visual comparison between NeRF, NeRD, and Neural-PIL: [Anonymous Link](https://www.dropbox.com/s/syut63hc2y3x5v7/NeuralPIL-Supplement-Addition.pdf?dl=0). This comparison shows renderings from novel view positions and also novel illuminations in cases of varying illumination scenes (Chair, Car, Globe, Gnome, Mother-Child). The following Table-R3 shows the metrics for all the scenes for NeRD and our model. We also included the LPIPS perceptual metric to further show that the results are also perceptually close to the GT. Both visual results and metrics show that we outperform NeRD in the majority of the scenes, while being faster to train and infer.

**Table-R3**: *Error on all individual scenes of the method. Including the perceptual error metric LPIPS.*:

| Type | Scene | NeRD-PSNR (Higher) | NeRD-SSIM (Higher) | NeRD-LPIPS (Lower) | Ours-PSNR (Higher) | Ours-SSIM (Higher) | Ours-LPIPS (Lower) |
|---|---|---|---|---|---|---|---|
| Real Varying | Gnome | 24.5862 | 0.8961 | 0.0915 | **25.9319** | **0.9291** | **0.0768** |
| Real Varying | Mother-Child | 25.9631 | 0.9602 | **0.0372** | **27.3636** | **0.9645** | 0.0616 |
| Real Fixed | Cape | 22.2830 | 0.8014 | 0.1933 | **23.2663** | **0.8484** | **0.1453** |
| Real Fixed | Head | 25.8802 | **0.9362** | **0.0550** | **25.9783** | 0.9342 | 0.0613 |
| Syn. Varying | Globe | 24.2559 | 0.9103 | 0.0721 | **24.8940** | **0.9229** | **0.0564** |
| Syn. Varying | Car | 29.7036 | 0.9667 | 0.0305 | **30.6387** | **0.9737** | **0.0308** |
| Syn. Varying | Chair | 29.7174 | 0.9609 | 0.0588 | **32.5741** | **0.9768** | **0.0345** |
| Syn. Fixed | Chair | **31.6864** | 0.9767 | **0.0272** | 31.4396 | **0.9801** | 0.0336 |
| Syn. Fixed | Lego | **30.3931** | **0.9709** | **0.0268** | 27.8887 | 0.9642 | 0.0426 |
| Syn. Fixed | Ship | **28.1087** | 0.9168 | 0.0876 | 26.4081 | **0.9180** | **0.0707** |

---

> ### Comment · Reviewer_rZtL · 2021-08-30
> **Thanks authors for the new results**
>
> Thanks authors for the new results, including the new results shown here and under my reviews. Given these results, I would like to change my rating to be positive. Authors results solve most of my questions and I like the idea using a learned manifold to constrain lighting estimation. If possible, I still hope that authors can include more than 2 real examples in the final version.

---

### Decision · Program_Chairs · 2021-09-27

**Decision:**

Accept (Poster)

**Comment:**

All reviewers were somewhat positive  on the paper after an extensive discussion.

They all agree on the novelty of the neural pre-integrated illumination network (Neural-PIL) as well as the advantages of using smooth manifold auto-encoders (SMAE).

Authors are strongly encouraged to include the new results they provided during rebuttal in the final version of the paper and to highlight their  method on real images.

Given the current interest in neural rendering, even though the visual improvements over the state of the art are somewhat subtle,  the Metareviewer thinks that the novel contributions of this paper should be present at NeurIPS.